# Look Ma, No GANs!
# Image transformation with ModifAE

## Abstract

Existing methods of image to image translation require multiple steps in the training or modification process, and suffer from either an inability to generalize, or long training times. These methods also focus on binary trait modification, ignoring continuous traits. To address these problems, we propose ModifAE: a novel standalone neural network, trained exclusively on an autoencoding task, that implicitly learns to make continuous trait image modifications. As a standalone image modification network, ModifAE requires fewer parameters and less time to train than existing models. We empirically show that ModifAE produces significantly more convincing and more consistent continuous face trait modifications than the previous state-of-the-art model.

## 1 Introduction

Image modification is a difficult task that requires changing some aspects of an image while keeping others static. These changes can include modifying the class of an image, often called image to image translation Isola et al. (2016), or they can refer to modifying a continuous trait in an image, which we will refer to as continuous image modification. These image modification tasks are often performed on datasets like CelebA because human faces have many easily recognizable traits Liu et al. (2015).

Initially, much of this work required paired examples or was done with hand-tailored methods, Kemelmacher-Shlizerman (2016); Laffont et al. (2014); Hertzmann et al. (2001); Khosla et al. (2013); Duong et al. (2016). This can be time consuming and can struggle with generalization. More general approaches have been introduced, but only for image to image translation. Some of these newer methods can modify continuous traits, such as the apparent age of a face, but they can only achieve this by breaking images into binary categories such as "young" and "old" or multiple age groups, such as 0-18, 19-29, and etc Upchurch et al. (2017); Gardner et al. (2015); Liu et al. (2017a); Antipov et al. (2017).

Another level of complexity is introduced when dealing with unpaired data Zhu et al. (2017). That is, given unrelated images from different domains, translate an image from one class to another. Without pairs of images in each domain, the problem cannot be solved as a simply mapping an input image to its paired output image.

The unpaired image to image translation task has been been tackled as a latent space traversal problem. For example, Upchurch et al. (2017) use deep features from a pretrained classifier to form a latent space in which different classes of images are loosely clustered. Then, by projecting an image into the latent space, interpolating the embedding towards the mean of a different class, and projecting back into the image space, class modifications can be achieved Upchurch et al. (2017); Gardner et al. (2015). However, these methods require multiple steps and use hand-tailored linear interpolations through a latent space. These methods take more time to design and do not generalize well to new domains. A more desirable method of modifying an image would do so based purely on learned weights, using a single forward pass of a neural network.

Over the past four years, generative adversarial networks Goodfellow et al. (2014) have become the most popular tool for the image to image translation task, allowing for the image modification to be fully controlled by the generator network in one forward pass. They use adversarial training to ensure results are believable and often use cycle training to ensure that identifying traits from the original image are preserved Zhu et al. (2017); Isola et al. (2016); Liu et al. (2017b); Choi et al. (2017);

Creswell et al. (2017); Liu et al. (2017a); Larsen et al. (2016); Perarnau et al. (2016). However, by definition, GANs require at least two networks to be trained. These networks also need to remain in equilibrium throughout the training process to prevent modal collapse Berthelot et al. (2017). A single-network image modification method would be better since it would have fewer parameters and train in a faster and simpler manner.

Additionally, none of these methods are built to work on continuous trait modification. They currently only use binary classes (old vs. young, or beard vs. clean shave). While it is possible to linearly interpolate between these binary classes, traits may not scale linearly between these classifications, e.g. aging between 10 to 15 years old would require a different change than aging from 30 to 35 years old.

As mentioned above, existing techniques either use hand-tailored methods, rely on multiple steps like projection and linear interpolation, or require two networks for training. To fill this gap, we introduce ModifAE: a general image modification method which works without GANs or hand-tailored features. It also works for continuous traits. To our knowledge, ModifAE is the first standalone neural network which can perform general image modification in a single forward pass.

## 2 RELATED WORK

### 2.1 AUTOENCODERS

One method of generating efficient encodings of images is the autoencoderHinton and Salakhutdinov (2006); Cottrell et al. (1987). These networks take in a high dimensional input, reduce to a lower dimensional representation, and then decode back to the original input. A bottleneck occurs in the middle of the autoencoder network, creating a latent space with high level features about the input. When the network is linear and has one hidden layer, they implement a version of PCA P. Baldi (1988). Additional hidden layers result in nonlinear encodings Hinton and Salakhutdinov (2006); DeMers and Cottrell (1993). More recently, variational autoencoders Kingma and Welling (2013) have been introduced. These networks create more continuous latent spaces where linear interpolations can be performed smoothly.

### 2.2 DEEP FEATURE INTERPOLATION

Some general methods of image to image translation are based on stepped linear traversals in a learned latent space. For example, Deep Feature Interpolation (DFI) Upchurch et al. (2017) relies on linear interpolation of the latent representation of an input image. DFI involves training a deep convolutional neural network Simonyan and Zisserman (2014); Krizhevsky et al. (2012) to classify images, then the network performs an image traversal based on a statically defined procedure, which is not learned by the network. Since DFI is designed to interpolate between means of class clusters, it cannot be used for continuous trait modification. Also, all modifications require a forward pass of the network, an interpolation in the latent space, then deconvolutions to arrive back at an image.

### 2.3 CONDITIONAL GENERATIVE ADVERSARIAL NETWORKS

Conditional and Controllable GANs Mirza and Osindero (2014); Lee and Seok (2017); Yin et al. (2017) made it possible to generate images conditioned on certain traits. The generator is trained with an image and trait input, then the discriminator gives feedback based on whether the image appears real or fake and whether it appears to belong to the target class Lee and Seok (2017). However, such networks can still struggle with preserving identifying traits of the original image.

Zhu et al. (2017) proposed **CycleGAN**, a model for image to image translation which addresses the challenge of maintaining identifying traits in the modified image. It enforces cycle consistency by modifying an image with a different target class, then modifying it back to its original classification. After the modification cycle, pixel-wise autoencoding loss is calculated for the original and reconstructed images. However, CycleGAN is incapable of modifying multiple traits with a single model.

Recently, **StarGAN** Choi et al. (2017) was introduced and beat other methods on image to image translation. It uses cycle training to preserve identity traits and can also modify multiple classes in the

same forward pass. Choi et al. (2017) found that their model produced superior quality modifications when simultaneously supervising multiple traits, achieving state-of-the-art performance.

These networks can perform image to image translations on multiple traits in a single forward pass of the generator, while maintaining identifying traits of the original image. However, as mentioned in the introduction, they are currently all built to modify binary class traits. They also all require training multiple networks in equilibrium, which takes more parameters and more time. ModifAE offers a solution to both of these issues.

# 3 METHODS

ModifAE is a single network, trained exclusively on an autoencoding task, that implicitly learns to modify perceived traits in images (illustrated in Figure 2). In this section, we discuss the dataset, ModifAE architecture, training procedure, and how ModifAE works.

## 3.1 CONSTRUCTING A CONTINUOUS TRAIT DATASET

To evaluate ModifAE on continuous continuous face traits, we need a large dataset. We use images from the CelebA datasetLiu et al. (2015), which consists of over 200,000 images of celebrities (including, ironically, B-list celebrities!). The images in CelebA are annotated with 40 facial traits, such as "mustache", "eyeglasses", "hat", etc. Given that these are binary traits, rather than continuous ones, these labels are not appropriate for continuous trat testing.

To generate continuous traits of these faces, we use a system that was trained to "see people like people" Song et al. (2017) to rate faces from the CelebA dataset on continuous traits. Bainbridge et al. (2013) had workers judge 40 social traits of 2,222 faces from the the MIT 10k US Adult Faces Database. These traits include trustworthiness, attractiveness, responsibility, aggressiveness, etc. The 2,222 labeled faces from the MIT Faces Database are not enough to train a robust generative model. Hence, we used a trained regression network to judge the social traits of 190,000 faces from the CelebA dataset Liu et al. (2015); Song et al. (2017) and use the predicted trait ratings to train ModifAE. Example faces and their predicted ratings are shown in Figure 1.

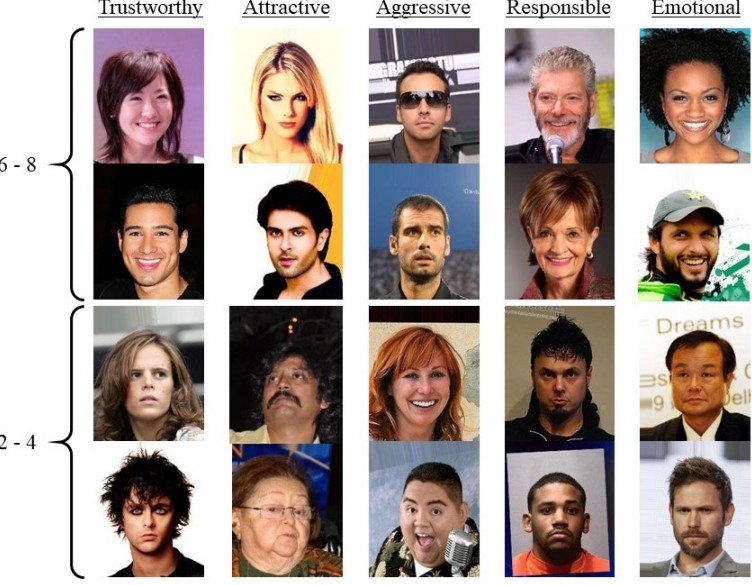

Figure 1: Examples of CelebA faces and their trait predictions.

To verify the effectiveness of this artificial dataset, we ran an AMT experiment checking how the predicted values align with human judgments in four attributes: aggressive, responsible, trustworthy

Table 1: AMT verification of our collected dataset

| Attribute | Chose "correct" member of the pair |
|---|---|
| Aggressive | 0.9509 |
| Emotional | 0.9234 |
| Responsible | 0.7783 |
| Trustworthy | 0.8780 |

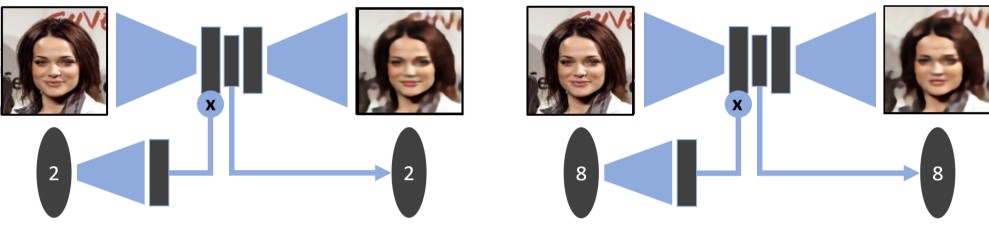

(a) Training with the real aggressiveness rating.   (b) Modifying with a desired aggressiveness rating.

Figure 2: Examples of ModifAE at training and usage times.

and emotional. Humans have high agreement in these traits. For each trait, we picked 40 images rated highest by the prediction network Song et al. (2017), and 40 images rated lowest. Next, we form 40 pairs by picking one image from the high rating group, and one image from the low group. We then ask AMT workers which face better exemplifies the corresponding trait in the each pair, (e.g. which one looks more trustworthy in this pair). Each trait's 40 pairs are evaluated by 30 workers. Then, we calculated the overall likelihood (among all the workers and among all the pairs) that the face of higher predicted score is chosen by human subjects for each attribute. As can be seen in Table 1, all the attributes predicted by the regression network Song et al. (2017) align well with human judgments. Therefore, we consider the predicted scores as being roughly equivalent to human judgments of those faces.

## 3.2 ARCHITECTURE

The ModifAE architecture consists of two autoencoding paths which fuse in the middle of the network. It is based on a typical convolutional image autoencoder architecture, specifically the discriminator autoencoder found in BEGAN Berthelot et al. (2017). As in the BEGAN discriminator, the image first goes through 14 convolutional layers, reducing the feature map size to 8x8. The differences lie in the bottleneck of ModifAE, where information about the traits to be applied to the image is fused with the image information.

The image autoencoding pathway is a fully convolutional network from input, through the bottleneck, to output. The trait autoencoding path consists of a fully connected (FC) layer from the input to the image path's bottleneck, then another FC layer from the bottleneck back to the traits (Figure 3).

The middle three layers of the image path form the bottleneck where the trait encoding is fused with the image encoding. At the first layer of the bottleneck, the image feature map is $8\times8$ with 256 channels with a linear activation function. In order to combine the image information with the trait information, we train two FC layers to project the traits to a 10-dimensional embedding space, and then to the current image feature map size. Then, the values in the image feature maps are multiplied[1] by the projected trait values.

We encourage a combined representation at this point by constraining the bottleneck further in the "fuse layer." This layer reduces the bottleneck to 16 filters, using $1\times1$ convolutions, fusing together the image and trait information. Then, those values are used to predict the trait output through another FC layer with linear activation. This concludes the trait pathway, and the image pathway

---

[1]We achieved similar results by adding trait activations to the latent space, but generated results for this paper with multiplication.

continues symmetrically from these feature maps to arrive back at original dimensions on output. The architecture is depicted in Figure 3.

While training our model, we use the known values for each trait as input, facilitating a learned shared latent space through the autoencoding loss. However, during test time, the input trait values can be any desired value for the modification. Because the output image passes through the fused embedding space with the traits, changing the traits results in a modified output image.

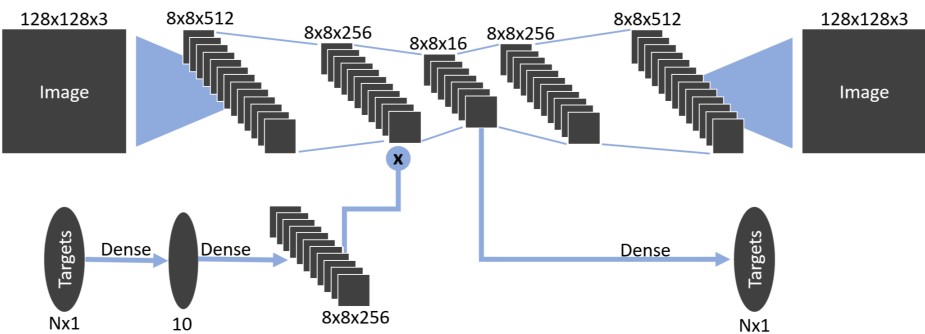

Figure 3: General illustration of ModifAE architecture.

### 3.3 TRAINING

ModifAE is only trained on an autoencoding task. We train ModifAE using the Adam optimizer Kingma and Ba (2014) and train for 15 epochs on CelebA images Liu et al. (2015). The objective is to optimize a single loss function based on two terms. Following BEGAN Berthelot et al. (2017), we use the $L_1$ loss on the image autoencoder. We also optimize the $L_1$ trait autoencoding loss. The total loss is:

$$L = \frac{1}{N} \sum_{p=1}^{N} |x_p - AE(x_p)| + |y_p - AE(y_p)| \tag{1}$$

where $x_p$ is the $p^{th}$ image example, $y_p$ is its trait vector and $AE(\cdot)$ is the output of the autoencoder on each. At training time, the objective is to take in an image and its traits, autoencode both, and arrive at outputs identical to the inputs. No form of adversarial or cycle training is necessary. Despite this, the trained network can modify images without obscuring identity traits.

### 3.4 WHY IT WORKS

The first half of the image path encodes the image down to a bottlenecked latent space, where higher level features about the image are encoded. Since the supervised traits refer to some of these high-level features, we decided to fuse the image and trait information here.

By scaling the projection of the image in the latent space by the traits at the image bottleneck, we interfere with the image autoencoding process. The network could minimize the pixel-wise image autoencoding error by learning to always map the trait projection to one in the latent space, except part of the objective function is to reproduce the original traits on output. Therefore, the network is forced to intelligently scale values in the latent space, preserving trait information without adding noise to the image.

The model achieves this by projecting both the image and the the traits to some shared latent space. In this shared latent space, the image is encoded to express all targets with some combination of feature maps, and the trait pathway learns connections to scale the latent image feature maps corresponding to the trait. Once the image and traits are fused in the latent space, changing the input traits will result in a reasonably edited output image, where only the features relevant to the target traits are changed.

# 4 RESULTS

Here, we provide examples of ModifAE's performance on the novel continuous image modification task. We qualitatively compare ModifAE's and StarGAN's results for the same task, quantitatively compare the modification effectiveness with a user study, and numerically compare the ModifAE architecture with recent image to image translation methods.

## 4.1 QUALITATIVE RESULTS ON CONTINUOUS MODIFICATIONS

### 4.1.1 MULTI-TRAIT TRAVERSALS

As emphasized by Choi et al. (2017), the ability to modify multiple traits with a single model is important. Here, we show that ModifAE is capable of making continuous modifications on multiple traits (see Figure 4). For this experiment, we trained ModifAE on two traits: "attractive" and "emotional." The picture in the upper left corner is the original, with its true trait values next to it. Looking at the (0,0) point in the woman's results (unattractive and unemotional) her prominent cheekbone appears to sag, and her smile becomes a frown. In general, as she becomes more emotional, her smile increases, and as she is made more attractive, her smile increases as well, as smiling subjects are judged as more attractive.

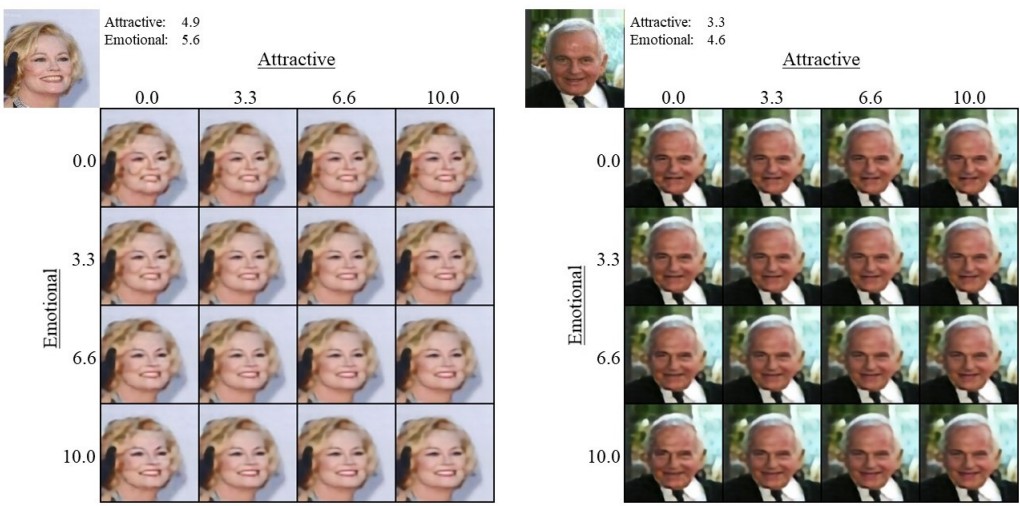

Figure 4: Continuous value, multi-trait image modifications by ModifAE.

### 4.1.2 QUALITATIVE COMPARISON TO STARGAN

To compare our model to StarGAN Choi et al. (2017), we binarize the continuous traits by doing a median split on the continuous-valued traits and trained StarGAN on these two groups (low and high). The results are shown in Figure 5. StarGAN's generator architecture has a larger bottleneck and includes residual layers, so the model easily produces higher resolution images than ModifAE. However, the faithfulness of the modification of the two models is worth comparing.

### 4.1.3 QUANTITATIVE COMPARISON TO STARGAN

To verify the quality of these continuous subjective trait modifications, we assessed human interpretations of the faces. For this, we perform an AMT experiment where we consider two traits: "trustworthy" and "aggressive." For these traits, we present participants with a sequence of 120 image pairs. They are asked to pick which image most exemplifies the trait.

Each sequence contains 10 ground truth pairs and 110 modified pairs. Each subject was randomly assigned to either ModifAE- or Stargan-modified images, in order to avoid judgments influenced by image resolution. A "ground truth" pair denotes a pair in which both images are the original

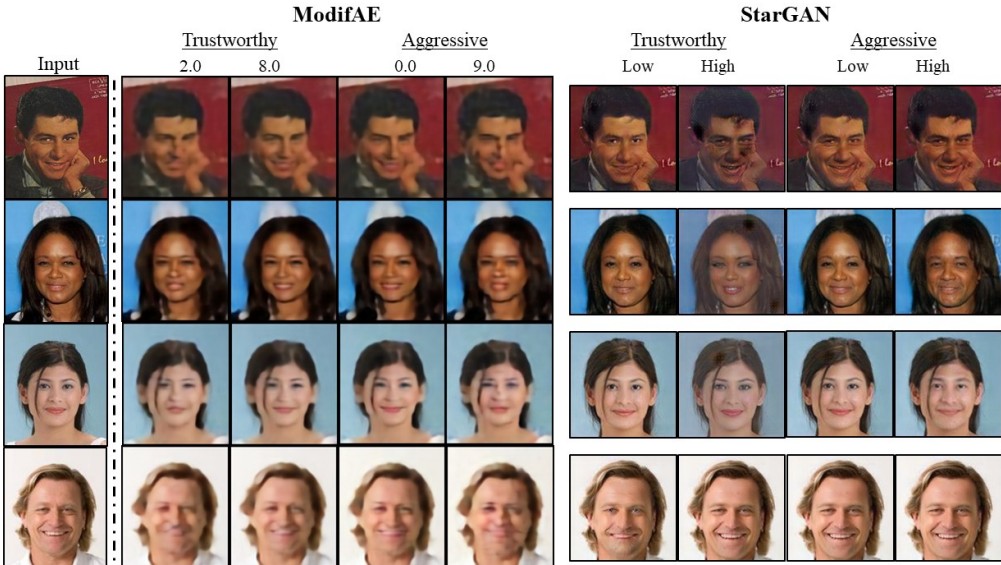

Figure 5: Comparison of ModifAE and StarGAN modifications.

Table 2: Human evaluation of modified images

| Attribute | Performance | ModifAE | StarGAN |
|---|---|---|---|
| Aggressive | Ground Truth | 0.9363 | 0.9546 |
| | Same Face (S) | 0.8382 | 0.8203 |
| | Different Faces (D) | 0.6587 | 0.5916 |
| | Average of S and D | 0.7484 | 0.7059 |
| | | | |
| Trustworthy | Ground Truth | 0.8571 | 0.8833 |
| | Same Face (S) | **0.7962** | 0.3898 |
| | Different Faces (D) | 0.5956 | 0.5228 |
| | Average of S and D | **0.6959** | 0.4563 |

unmodified images from the CelebA dataset, and these pairs were used to verify that workers were paying attention to the task. Since we have validated that our predicted trait scores align well with human judgment, we know the putative "correct answers" for these pairs. The overall ground truth results are between 85 and 96%, so we didn't discard any subjects.

For the modified pairs, we generated two types of pairs: (1) same-face pairs, and (2) different-face pairs. The same-face pairs are modifications of the same photo, one in which the target trait is increased, and one in which it is decreased. To make a different-face pair, photos of two different people with similar scores in the target trait are chosen and then one is modified to increase the trait, and the other is modified to decrease the trait. Both the same-face and different-face sets contain 45 pairs in which no image is repeated. However, to verify the workers' consistency, 20 of the 90 modified pairs were repeated to result in 110 modified pairs.

We calculate the fraction of subjects that chose the image with the increased trait. This is shown in Table 2. This can be used as a proxy for performance because higher values indicate more human agreement with the model's modifications. To verify the data collected in each model's experiment, we performed a group t-test and found no statistical differences between the ground truth performance in the ModifAE and StarGAN experiments. This shows that the results between models are not due to an inequality in workers.

For modifications on "aggressive," ModifAE received higher scores for both same-face and different-face pairs, but the difference is not statistically significant. However, for modification on "trust-

Table 3: Model size for learning seven traits

| Model | Parameters |
| --- | --- |
| CycleGAN | 736M |
| ICGAN | 68M |
| StarGAN | 53M |
| **ModifAE** | **24M** |

worthy", there are significant differences in the overall average performance and same-face pairs performance (p<0.001). By examining the distribution closely, we find that StarGAN is less consistent. Some StarGAN pairs receive high scores, while many other pairs cause most workers to agree on the wrong image. Meanwhile, for ModifAE, most pairs get at the majority vote for the correct image. We also observe that people are more consistent with ModifAE generated faces. On the 20 repeated pairs, 91.6% of the choices are the same for ModifAE, where for StarGAN, the number is only 45.0%.

## 4.2 MODEL SIZE

A novel aspect of ModifAE is its ability to modify images in a single forward pass, based purely upon learned weights, despite only having one network. Other models that can modify images in a single forward pass are GANs. By comparison, ModifAE, as a single neural network, requires fewer parameters and less time to train. While StarGAN takes one day to train on CelebA images Choi et al. (2017), ModifAE takes less than 10 hours. Table 3 shows the number of parameters required by different models training on seven traits in the CelebA dataset. The listed values are as reported in the original papers and in the parameter comparisons of Choi et al. (2017) Perarnau et al. (2016); Zhu et al. (2017).

## 5 CONCLUSION

In this paper, we propose ModifAE: a novel image modification network that can edit continuous traits in a single forward pass, based solely on learned weights. ModifAE does not require training multiple networks or performing multiple steps for image modification. Instead, a single network is trained to autoencode an image and its traits through the same latent space, implicitly learning to make meaningful changes to images based on trait values. Our experiments show that ModifAE requires fewer parameters and takes less training time than existing forward-pass methods Zhu et al. (2017); Perarnau et al. (2016); Choi et al. (2017). In addition, we computed and verified novel continuous subjective trait ratings for CelebA faces. We will make this augmentation to the CelebA dataset available upon paper acceptance. Finally, we demonstrated that ModifAE makes more meaningful continuous image traversals than equivalents generated with a state-of-the-art method Choi et al. (2017). Theoretically, ModifAE is not limited to face trait modification, and the resolution of output images can be greatly increased. ModifAE shows great promise as an easy-to-train model for the general task of image modification.

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
