# OpenReview forum: "Look Ma, No GANs! Image Transformation with ModifAE"
_ICLR.cc/2019/Conference_

### Official Review · AnonReviewer3 · 2018-11-03
**an autocoding model as an alternative to GANs for continuous trait image modifications**

**Rating:** 5
**Confidence:** 3

**Review:**

Overview and contributions: The authors propose the ModifAE model that is based on an autoencoder neural network for continuous trait image modifications. ModifAE requires fewer parameters and less time to train than existing generative models. The authors also present experiments to show that ModifAE produces more convincing and more consistent continuous face trait modifications than the current baselines.

Strengths:
1. Nice presentation of the model.
2. Good experiments to justify improved running time and fewer number of parameters.

Weaknesses:
1. I am not completely convinced by the results in Figure 4. It doesn't seem like the model is able to pick up on subtle facial expressions and generate them in a flexible manner. In fact, the images look very very similar regardless of the value of the traits. Furthermore, the authors claim that "In general, as she becomes more emotional, her smile increases, and as she is made more attractive, her smile increases as well, as smiling subjects are judged as more attractive". I believe attractiveness and emotions are much more diverse and idiosyncratic than just the size of her smile...
2. From Figure 5 it seems like ModifAE generates images that are lower in quality as compared to StarGAN. Can the authors comment on this point? How can ModifAE be improved to generate higher-quality images?

Questions to authors:
1. Weakness points 1 and 2.
2. This did not affect my rating, but I am slightly concerned by the labelings as seen in Figure 1. Is it reasonable to infer traits like "trustworthy", "attractive", "aggressive", "responsible" from images? Are these traits really what we should be classifying people's faces as, and are there any possible undesirable/sensitive biases from the dataset that our models could learn? I would like to hear the author's opinions on the ethical implications of these datasets and models.

Presentation improvements, typos, edits, style, missing references:
None

---

### Official Review · AnonReviewer2 · 2018-11-04
**Modify social attributes on face images results here in low-quality images**

**Rating:** 4
**Confidence:** 4

**Review:**

The paper is about changing the attributes of a face image to let it look more aggressive, trustworthy etc. by means of a standalone autoencoder (named ModifAE). The approach is weak starting from the construction of the training set. Since continue social attributes on face images does not exist yet, CelebA dataset is judged by Song et al. (2017) with continuous face ratings and use the predicted ratings to train ModifAE. This obviously introduces a bias driven by the source regression model. The hourglass model is clearly explained. The experiments are not communicating: the to qualitative examples are not the best showcase for the attributes into play (attractive, emotional), and requires to severely magnify the pdf to spot something. This obviously show the Achille’s heel of these works, i.e., working with miniature images. Figure 5, personally, is about who among modifAE and stargan does less bad, since the resulting images are of low quality (the last row speaks loud about that)
Quantitative results are really speaking user tests, so I will cal it as they are, user tests. They work only on two attributes, and show a reasonable advantage over stargan only for one attribute.

---

### Official Review · AnonReviewer4 · 2018-11-10
**Authors propose a trait modification network that is trained as a standalone auto-encoder and is able to model continous trait interpolations in the latent space. The network requires less parameters than competitors and the training process less painful than other approaches based on GANs.**

**Rating:** 3
**Confidence:** 4

**Review:**

* The construction of the training dataset is clearly flawed by the use of an automatic algorithm that would certainly introduce a strong bias and noisy labels. Even though the dataset is supposed to encode continuous traits, the validation with human subjects is performed in a binary fashion.

* I miss more formality in the presentation of the methodology. Figure 3. does not seem very self-explanatory, nor does the caption. Which is the dimensionality of the input trait vector?. I assume the input would be the trait ratings predicted by the human subjects. However in the experiments training seems to be done with a maximum of two traits. This makes me wonder how the dense part of the network can handle the dimensionality blow-up to match the latent space dimensionality without suffering from overfitting. I would appreciate some disussion regarding this.

* While I appreciate a section reasoning why the method is supposed to work, those claims should be backed with an ablation study in the experimental section.

* The qualitative results show a few examples which I find very hard to evaluate due to the low-resolution of the predictions. In both traits there seems to be the same facial features modified and I can't find much difference between trustworthy and aggresssive (the labels could be swapped and I would have the same opinion on the results). I miss additional trait examples that would make clearer if the network is learning something besides generating serious and happy faces.

* The qualitative comparison with StarGAN seems unfair, as if one checks the original paper their results are certainly more impressive than what Figure 5 shows.

* The authors show only two traits in the experiments which makes me a bit suspicious about the performance of the network with the rest of traits. The training datset considers up to 40 traits.

---

### Meta-Review · Area_Chair1 · 2018-12-13
**interesting problem, but experimental results aren't promising**

**Confidence:** 4
**Recommendation:** Reject

**Metareview:**

1. Describe the strengths of the paper.  As pointed out by the reviewers and based on your expert opinion.

- The paper tackles an interesting and relevant problem for ICLR: guided image modification of images (in this case of facial attributes).
- The proposed method is in general well-explained (although some details are lacking)

2. Describe the weaknesses of the paper. As pointed out by the reviewers and based on your expert opinion. Be sure to indicate which weaknesses are seen as salient for the decision (i.e., potential critical flaws), as opposed to weaknesses that the authors can likely fix in a revision.

- The training set of faces and associated attributes were annotated using a pre-trained model which introduced a bias into the annotations used for training the method.
- The experimental results weren't convincing. The qualitative results showed no clear advantage of the proposed method and the quantitative comparison to StarGAN only considered two attribute manipulations and only found a statistically significant different in performance for one of those.
The second weakness was the key determining factor in the AC's final recommendation.

3. Discuss any major points of contention. As raised by the authors or reviewers in the discussion, and how these might have influenced the decision. If the authors provide a rebuttal to a potential reviewer concern, it’s a good idea to acknowledge this and note whether it influenced the final decision or not. This makes sure that author responses are addressed adequately.

There were no major points of contention and no author feedback.

4. If consensus was reached, say so. Otherwise, explain what the source of reviewer disagreement was and why the decision on the paper aligns with one set of reviewers or another.

The reviewers reached a consensus that the paper should be rejected.